# Impact of Amerind ancestry and FADS genetic variation on omega-3 deficiency and cardiometabolic traits in Hispanic populations

Chaojie Yang [1,2], Brian Hallmark [3], Jin Choul Chai[4], Timothy D. O'Connor [5], Lindsay M. Reynolds [6], Alexis C. Wood[7], Michael Seeds[8], Yii-Der Ida Chen[9], Lyn M. Steffen[10], Michael Y. Tsai[11], Robert C. Kaplan[4,12], Martha L. Daviglus[13], Lawrence J. Mandarino[14], Amanda M. Fretts[15], Rozenn N. Lemaitre [16], Dawn K. Coletta [14,17], Sarah A. Blomquist[18], Laurel M. Johnstone[19], Chandra Tontsch[18], Qibin Qi[4], Ingo Ruczinski[20], Stephen S. Rich [1], Rasika A. Mathias [21], Floyd H. Chilton[18,22] & Ani Manichaikul [1,22 ✉]

Long chain polyunsaturated fatty acids (LC-PUFAs) have critical signaling roles that regulate dyslipidemia and inflammation. Genetic variation in the *FADS* gene cluster accounts for a large portion of interindividual differences in circulating and tissue levels of LC-PUFAs, with the genotypes most strongly predictive of low LC-PUFA levels at strikingly higher frequencies in Amerind ancestry populations. In this study, we examined relationships between genetic ancestry and *FADS* variation in 1102 Hispanic American participants from the Multi-Ethnic Study of Atherosclerosis. We demonstrate strong negative associations between Amerind genetic ancestry and LC-PUFA levels. The *FADS* rs174537 single nucleotide polymorphism (SNP) accounted for much of the AI ancestry effect on LC-PUFAs, especially for low levels of n-3 LC-PUFAs. Rs174537 was also strongly associated with several metabolic, inflammatory and anthropomorphic traits including circulating triglycerides (TGs) and E-selectin in MESA Hispanics. Our study demonstrates that Amerind ancestry provides a useful and readily available tool to identify individuals most likely to have *FADS*-related n-3 LC-PUFA deficiencies and associated cardiovascular risk.

[1] Center for Public Health Genomics, University of Virginia, Charlottesville, VA, USA. [2] Department of Biochemistry and Molecular Genetics, University of Virginia, Charlottesville, VA, USA. [3] Center for Biomedical Informatics and Biostatistics, University of Arizona, Tucson, AZ, USA. [4] Department of Epidemiology and Population Health, Albert Einstein College of Medicine, Bronx, NY, USA. [5] Institute for Genome Sciences; Program in Personalized and Genomic Medicine; Department of Medicine, University of Maryland School of Medicine, Baltimore, MD, USA. [6] Department of Epidemiology and Prevention, Division of Public Health Sciences, Wake School of Medicine, Winston-Salem, NC, USA. [7] USDA/ARS Children's Nutrition Research Center, Baylor College of Medicine, Houston, TX, USA. [8] Molecular Medicine, Wake Forest University, Winston-Salem, NC, USA. [9] Institute for Translational Genomics and Population Sciences and Department of Pediatrics, Los Angeles Biomedical Research Institute at Harbor-UCLA Medical Center, Torrance, CA, USA. [10] Division of Epidemiology and Community Health, University of Minnesota School of Public Health, Minneapolis, MN, USA. [11] Department of Laboratory Medicine and Pathology, University of Minnesota, Minneapolis, MN, USA. [12] Division of Public Health Sciences, Fred Hutchinson Cancer Research Center, Seattle, WA, USA. [13] Institute for Minority Health Research, University of Illinois at Chicago, Chicago, IL, USA. [14] Department of Medicine, Division of Endocrinology, University of Arizona College of Medicine, Tucson, AZ, USA. [15] Department of Epidemiology, Cardiovascular Health Research Unit, University of Washington, Seattle, WA, USA. [16] Department of Medicine, Cardiovascular Health Research Unit, University of Washington, Seattle, WA, USA. [17] Department of Physiology, University of Arizona College of Medicine, Tucson, AZ, USA. [18] Department of Nutritional Sciences, University of Arizona College of Agriculture and Life Sciences, Tucson, AZ, USA. [19] University of Arizona Genetics Core, University of Arizona, Tucson, AZ, USA. [20] Department of Biostatistics, Johns Hopkins University, Baltimore, MD, USA. [21] Department of Medicine, Johns Hopkins University, Baltimore, MD, USA. [22] These authors jointly supervised this work: Floyd H. Chilton, Ani Manichaikul. ✉email: amanicha@virginia.edu

Human diets in developed countries have changed dramatically over the past 75 years, leading to increased obesity, inflammation, cardiometabolic disorders, and cancer risk, possibly due to interactions between genotype with diet and other factors. Certain racial/ethnic groups carry a disproportionate burden of preventable negative outcomes and associated mortality[1–3]. Hispanic populations represent the largest racial/ethnic US minority where, compared to non-Hispanic whites, they have higher rates of obesity[4], poorly controlled high blood pressure[5], and elevated circulating triglycerides (TGs)[6], Hispanic populations also demonstrate a higher prevalence of diabetes and nonalcoholic fatty liver disease (NAFLD) than other racial/ethnic populations in the United States[7,8]. Hispanic Americans represent a heterogenous group with respect to ancestry, with notable differences in cultural/ lifestyle factors and disease prevalence based on country of origin. In particular, Hispanics identifying with the higher Amerind (AI)-ancestry origin have demonstrated enhanced urine albumin excretion[9], heart failure[10], lupus erythematosus risk[11], and prevalence of NAFLD compared to other Hispanic populations[12], supporting the critical need to conduct studies in these large, rapidly growing populations.

Omega-3 (n-3) and omega-6 (n-6) long chain (20–22 carbon; LC-) polyunsaturated fatty acids (PUFAs) and their metabolites play vital roles in innate immunity, energy homeostasis, brain development, and cognitive function[13–19]. LC-PUFAs are critical signaling molecules for immunity and inflammation with most evidence showing that n-3 and n-6 LC-PUFAs and their metabolic products have different and often opposing effects[20–24]. Metabolites of the n-6 LC-PUFA arachidonic acid (ARA) typically act locally to promote inflammatory responses[25–27], while n-3 LC-PUFAs, such as eicosapentaenoic acid (EPA) and docosahexaenoic acid (DHA) and their metabolites, have anti-inflammatory and pro-resolution properties (meaning that they promote resolution of inflammation)[28,29]. In addition to their effects on inflammation, circulating levels of n-3 LC-PUFAs, including EPA and DHA, are inversely associated with fasting and postprandial serum TG concentrations, largely through attenuation of hepatic very-low-density lipoprotein (VLDL)-TG production[30,31]. Dietary supplementation with these n-3 LC-PUFAs has been shown consistently to reduce fasting circulating TG levels and improve lipid accumulation associated with NAFLD[32,33].

The biosynthesis of n-3 and n-6 LC-PUFAs transpires via alternating desaturation (Δ6, Δ5, and Δ4) and elongation enzymatic steps encoded by fatty acid desaturase (FADS) cluster genes (FADS1 and FADS2), and fatty acid elongase genes (ELOVL2 and ELOVL5), and there is a limited capacity for biosynthesis through this pathway[34–36]. As a result, the primary dietary PUFAs that enter this pathway (linoleic acid [18:2n-6; LA], α-linolenic acid [18:3n-3; ALA], and their metabolic intermediates) compete as substrates for the desaturation and elongation steps. Additionally, early studies with deuterated substrates indicated there is a saturation point where additional dietary quantities of 18 carbon dietary substrates had no effect on circulating LC-PUFA levels[37]. These studies also estimated that conversion of dietary ALA provided 75–85% of total n-3 LC-PUFAs needed to meet daily requirements[37].

In 1961, a major effort was initiated to reduce levels of saturated fatty acids and replace them with PUFAs in an attempt to reduce circulating LDL-cholesterol and TGs[38–40]. This in turn led to a dramatic increase in eighteen carbon (18C-) PUFA-containing vegetable oils such as soybean, corn, and canola oils that contain high levels of n-6 LA relative n-3 ALA. It has been estimated that dietary LA increased from 2.79 to 7.21% of energy, whereas there was only a modest elevation in ingested ALA (from 0.39 to 0.72%), resulting in a ~15:1 ratio of LA to ALA entering

the LC-PUFA biosynthetic pathway and an estimated 40% reduction in total circulating n-3 LC-PUFA levels[41]. Since LA and ALA compete for the same desaturation and elongation steps and there is a limited capacity for n-6 and n-3 LC-PUFA biosynthesis through the pathway, several human and animal studies suggested that the dramatic shift in quantities and ratios of dietary LA and ALA could lead to imbalances in n-6 to n-3 LC-PUFAs and, potentially, n-3 LC-PUFA deficiencies[42–46] Thus, as certain populations moved from traditional to modern Western diets (MWD), it was suggested excess LA would lead to "omega-3 deficiency syndrome"[47].

The rate-limiting step of LC-PUFA biosynthesis has long been recognized to be the FADS-encoded Δ6 and Δ5 desaturation steps. Over the past decade, GWAS and candidate gene studies have shown that variation in the FADS gene locus on human chromosome 11 is strongly associated with plasma levels of ARA and EPA and the efficiency by which LC-PUFA precursors (18C dietary PUFAs) are metabolized to n-6 and n-3 LC-PUFAs[48,49]. FADS cluster genetic variation is associated with numerous molecular phenotypes that impact human disease as well as the risk of several diseases, including coronary heart disease[50], diabetes[51–53], and colorectal cancer[54]. FADS cluster genetic variation is strongly associated with circulating TG and VLDL concentrations in young healthy Mexicans[55].

Our previous studies revealed that African (compared to European) ancestry populations had elevated levels of LC-PUFAs, an increased frequency of the associated FADS genetic variants and a more efficient LC-PUFA biosynthesis (termed the derived haplotype)[56]. In contrast, FADS variants associated with more limited capacity to synthesize LC-PUFAs (termed ancestral haplotype) are nearly fixed in Native American and Greenland Inuit populations and found at high frequencies in Amerind (AI) Ancestry Hispanic populations[56]. These distinct patterns of haplotypes have resulted in part from positive selection for the ancestral haplotype among Indigenous American populations[56].

While the role of FADS variation in modulating circulating fatty acid levels has been documented previously[48,49], prior studies have not examined the impact that population differences in FADS allele frequencies have in downstream population-specific risk of fatty acid deficiency, The hypothesis tested in this paper is that ancestral FADS variation in the context of MWD is associated with low (perhaps inadequate) circulating levels of LC-PUFAs (particularly n-3 LC-PUFAs) in a large proportion of high AI-Ancestry Hispanic populations compared to other Hispanic populations, with downstream effects on numerous cardiometabolic and inflammatory risk factors. To address this question, we first examined the relationship between the genomic proportions of AI ancestry and circulating phospholipid LC-PUFA levels in self-reported Hispanic individuals from the Multi-Ethnic Study of Atherosclerosis (MESA)[57], which includes Hispanic groups with varying levels of AI ancestry. Second, we assessed the extent to which this relationship is explained by genetic variation within the FADS1/2 locus, and also examined the impact of FADS genetic variation on cardiometabolic and inflammatory risk factors (lipids, anthropometric, and inflammatory markers). Third, we tested whether these FADS genetic associations replicated in two high AI-Ancestry Hispanic cohorts, the Arizona Insulin Resistance (AIR) Registry[58], and the Hispanic Community Health Study/Study of Latinos (HCHS/SOL)[59,60].

## Results

**Participant characteristics**. The MESA participants[57,61] included in this analysis comprised 1102 unrelated individuals aged 45–84 years at baseline of self-reported Hispanic race/ethnicity with country-specific classification based on the birthplace of parents

and grandparents corresponding to Central American ($n = 80$), Cuban ($n = 45$), Dominican ($n = 145$), Mexican ($n = 572$), Puerto Rican ($n = 167$) and South American ($n = 93$) (Table 1). MESA Hispanic participants were recruited primarily from three field centers in the United States (Columbia University, University of California—Los Angeles (UCLA), and the University of Minnesota). The global proportions of AI, African, and European genetic ancestry in each individual were estimated using genome-wide SNP data (Table 1). Higher frequencies of the rs174537 T allele in the *FADS* cluster (corresponding to the ancestral allele) were observed in subjects with country/region-specific origins in Central America (0.59), South America (0.56), and Mexico (0.59) compared to those of Dominican (0.27), Cuban (0.28), or Puerto Rican origin (0.40) (Table 1).

**LC-PUFA levels are associated with Amerind genetic ancestry.** Higher proportions of AI genomic ancestry were associated with lower levels of LC-PUFAs in MESA Hispanics participants. Figure 1a, c and e shows levels of EPA, DHA, and ARA (expressed as the percentage of total fatty acids here and throughout the entire manuscript) as a function of inferred AI ancestry. Overall, AI ancestry explained 12.32%, 12.30%, and 12.48% of total variation in EPA, DHA, and ARA, respectively. Each 10% increase in AI ancestry was associated with a decrease of EPA (0.049), DHA (0.185), and ARA (0.401) in phospholipids. Between subjects with the lowest and highest proportions of AI ancestry, the n-3 LC-PUFAs decreased by 60.6% (for EPA) and 46.8% (for DHA) and the n-6 LC-PUFAs decreased by 30.7% (for ARA). Consequently, the nadir in predicted fatty acids levels in plasma phospholipids among those with 100% AI ancestry was ~0.3 and ~2 for EPA and DHA, respectively, compared to ~8.6 for the n-6 LC-PUFA, ARA.

Given the prior evidence that key genetic determinants of LC-PUFAs mapping to the *FADS* locus show strong variation in frequency between populations, we sought to determine the role of *FADS* variation in the relationships between LC-PUFA levels and global AI ancestry. LC-PUFAs were adjusted for rs174537 genotype (Fig. 1b, d and f); rs174537 is selected as a representative proxy SNP for the well-documented associations between the *FADS* locus and LC-PUFAs[62,63]. The rs174537 SNP has a strong effect on the ancestry-related decline in all LC-PUFAs. After adjusting for rs174537 genotype, an inverse association remains between global proportion of AI ancestry and EPA ($\beta = -0.30$, 95% confidence interval [Ci] = [$-0.39$, $-0.22$], $P = 9.05 \times 10^{-12}$ calculated using a two-sided *t*-test for the regression coefficient derived with $n = 1102$), DHA ($\beta = -1.42$, 95% CI = [$-1.76$, $-1.08$], $P = 6.76 \times 10^{-16}$ calculated using a two-sided *t*-test for the regression coefficient derived with $n = 1102$) and ARA ($\beta = -0.99$, 95% CI = [$-1.59$, $-0.38$], $P = 0.0015$ calculated using a two-sided *t*-test for the regression coefficient derived with $n = 1102$). Regression analysis of n-3 and n-6 LC-PUFAs with global proportion of AI ancestry, accounting for covariates: age, sex, and fish intake (Model 1), resulted in inverse relationships between the global proportion of AI ancestry with EPA ($\beta = -0.48$, $P = 3.7 \times 10^{-23}$ calculated by a Z-test from inverse variance weighted meta-analysis with a total of $n = 1057$), DPA ($\beta = -0.18$, $P = 7.6 \times 10^{-6}$ based on a Z-test from inverse variance weighted meta-analysis with a total of $n = 1057$), DHA ($\beta = -0.63$, $P = 0.0007$ based on a Z-test from inverse variance weighted meta-analysis with a total of $n = 1057$) and ARA ($\beta = -4.06$, $P = 1.3 \times 10^{-16}$ based on a Z-test from inverse variance weighted meta-analysis with a total of $n = 1057$) (Supplementary Table 1). These effects were consistent across study sites in MESA, with the largest effects observed at the University of Minnesota field center (Supplementary Table 1). Accounting for rs174537 genotype (Model 2), there remained an inverse association between the

global proportion of AI ancestry with EPA ($\beta = -0.28$, $P = 3.7 \times 10^{-08}$ based on a Z-test from inverse variance weighted meta-analysis with a total of $n = 1057$) (Supplementary Table 1), while the relationship of global proportion of AI ancestry with ARA, DPA, and DHA was no longer significant (Supplementary Table 1). In a model further accounting for local AI ancestry in addition to rs174537 (Model 3), EPA continued to be inversely associated with global proportion of AI ancestry ($\beta = -0.34$, $P = 8.4 \times 10^{-07}$ based on a Z-test from inverse variance weighted meta-analysis with a total of $n = 1057$), while the associations with DPA, DHA, and ARA were not statistically significant (Supplementary Table 1). In additional analysis examining global and local ancestry as potential modifiers of the effect of rs174537 on circulating fatty acid levels, we did not observe statistically significant evidence of interaction (Supplementary Table 2).

As other studies have suggested different specific variants as potentially functional within the *FADS* region, we further repeated the analysis presented in Fig. 1 through sensitivity analysis focused on the *FADS* region variant rs174557 (Supplementary Fig. 1), a common variant that diminishes binding of *PATZ1*, a transcription factor conferring allele-specific down-regulation of *FADS1*[64]. After adjusting for rs174557 genotype, we observed association between global proportion of AI ancestry and LC-PUFA levels (EPA: $\beta = -0.30$, $P = 2.19 \times 10^{-11}$; DHA: $\beta = -1.39$, $P = 2.54 \times 10^{-15}$; ARA: $\beta = -1.02$, $P = 0.0012$ calculated using a two-sided *t*-test for the regression coefficient derived with $n = 1102$) similar to that seen after adjusting for rs174537.

**Association of global Amerind ancestry with triglycerides.** Higher global proportions of AI ancestry were significantly associated with higher levels of circulating triglycerides (TG) in MESA Hispanic participants ($\beta = 65.40$ mg/dL, 95% CI = [42.28, 88.52] $P = 3.58 \times 10^{-8}$ based on a two-sided *t*-test for the regression coefficient derived with $n = 1101$) (Fig. 2a). This relationship was attenuated after adjusting for rs174537 (Fig. 2b), although there remained a significant relationship between global proportions of AI ancestry and TG levels ($\beta = 39.47$ mg/dL, 95% CI = ([2.62, 52.05], $P = 8.16 \times 10^{-04}$ based on a two-sided *t*-test for the regression coefficient derived with n = 1101). In sensitivity analysis, circulating triglycerides (TG) were adjusted for the variant rs174557. The relationship between global proportion of AI ancestry and TG levels (Supplementary Fig. 2; $\beta = 38.93$ mg/dL, $P = 9.59 \times 10^{-04}$ based on a two-sided *t*-test for the regression coefficient derived with $n = 1101$) is similar with the association adjusting for rs174537. Examining the unadjusted relationship between triglyceride levels and rs174537 genotype, we observed mean triglyceride levels increased with the number of copies of the rs174537 effect allele T (Fig. 3a). In analysis that incorporated adjustment for age and sex, the rs174537 T allele was significantly associated with higher levels of TG (GT vs GG: $\beta = 21.27$ mg/dL, 95% CI = [10.29, 32.25], $P = 0.0001$, TT vs GG: $\beta = 29.94$ mg/dL, 95% CI = [17.98, 41.88], $P = 1.01 \times 10^{-6}$ based on a two-sided *t*-test for the regression coefficient derived with $n = 1101$) (Table 2 and Fig. 3c).

**Association of PUFAs with *FADS* cluster SNPs.** We performed genetic association analysis adjusting for rs174537 genotype to determine if there was any residual association in the *FADS* region for the MESA Hispanic participants. In each of the Hispanic subgroups, after accounting for the rs174537 SNP, no additional genetic variants in the region were associated with EPA, DPA, DHA, or ARA (Supplementary Fig. 3). The rs174537 SNP is in strong linkage disequilibrium with other *FADS* cluster SNPs; thus, subsequent analyses are focused solely on the rs174537 SNP.

**Table 1 Participant characteristics for individuals of self-identified Hispanic origin from the MESA cohort.**

| Characteristics | Self-reported Hispanic country/region of origin | | | | | | | Total (N = 1102) |
|---|---|---|---|---|---|---|---|---|
| | Cuba (N = 45) | Dominican (N = 145) | Puerto Rico (N = 167) | South Amer. (N = 93) | Central Amer. (N = 80) | Mexico (N = 572) | | |
| Sex (Female) | 42.2% | 53.1% | 52.7% | 54.8% | 58.8% | 48.3% | | 50.6% |
| Age (years) | 69.8 (9.1) | 58.8 (10.1) | 59.3 (9.4) | 62.9 (10.3) | 58.7 (8.1) | 61.8 (10.2) | | 61.2 (10.1) |
| Study Site: Columbia University | 73.33% | 99.31% | 86.24% | 60.22% | 20% | 0.54% | | 35.93% |
| Study Site: University of Minnesota | 15.56% | 0.69% | 11.37% | 15.05% | 13.75% | 38.98% | | 24.95% |
| Study Site: UCLA | 11.11% | 0 | 2.39% | 24.73% | 66.25% | 60.48% | | 39.12% |
| Height (cm) | 163.1 (9.7) | 163.3 (9.4) | 162.6 (9.3) | 160.4 (8.9) | 159.6 (9.2) | 161.7 (9.5) | | 161.8 (9.4) |
| Weight (kg) | 75.3 (13.9) | 75.4 (14.3) | 79.5 (17.4) | 71.4 (12.3) | 74.9 (15.3) | 77.7 (15.8) | | 76.8 (15.6) |
| Waist-to-hip ratio | 0.98 (0.06) | 0.93 (0.08) | 0.94 (0.08) | 0.94 (0.07) | 0.96 (0.06) | 0.97 (0.07) | | 0.96 (0.07) |
| BMI (kg/m²) | 28.3 (5.3) | 28.2 (4.6) | 29.9 (5.7) | 27.7 (4.1) | 29.3 (5.2) | 29.6 (5.2) | | 29.3 (5.1) |
| HDL-C (mg/dl) | 49.8 (18.1) | 47.2 (10.7) | 49.7 (14.2) | 50.8 (13.9) | 48.0 (12.1) | 45.9 (12.5) | | 47.4 (13.0) |
| LDL-C (mg/dl) | 121.1 (26.1) | 124.7 (35.5) | 118.0 (33.3) | 115.8 (29.6) | 120.9 (38.3) | 119.4 (32.8) | | 119.8 (33.2) |
| Triglycerides (mg/dl) | 154.2 (100.4) | 132.9 (69.9) | 134.4 (72.1) | 151.5 (168.8) | 144.1 (75.5) | 173.6 (113.4) | | 157.5 (107.6) |
| s-ICAM (ng/ml) | 311.7 (71.8) | 262.4 (88.2) | 307.6(110.6) | 276.4 (72.3) | 286.6 (69.4) | 298.7 (80.6) | | 293.2 (86.2) |
| E-Selectin (ng/ml) | 64.05 (32.8) | 54.15 (17.9) | 63.65 (27.4) | 57.96 (29.8) | 62.74 (26.1) | 67.07 (29.3) | | 63.06 (27.4) |
| Fish intake (servings/day) | 0.19 (0.28) | 0.21 (0.22) | 0.21 (0.25) | 0.20 (0.28) | 0.22 (0.23) | 0.15 (0.21) | | 0.18 (0.23) |
| EPA (% of total fatty acids) | 0.90 (0.55) | 0.87 (0.68) | 0.76 (0.55) | 0.72 (0.43) | 0.58 (0.38) | 0.52 (0.29) | | 0.64 (0.46) |
| DPA (% of total fatty acids) | 0.98 (0.24) | 0.95 (0.26) | 0.90 (0.20) | 0.89 (0.22) | 0.83 (0.18) | 0.84 (0.19) | | 0.88 (0.21) |
| DHA (% of total fatty acids) | 3.71 (1.51) | 4.15 (1.31) | 3.58 (1.23) | 3.76 (1.25) | 3.24 (1.15) | 2.69 (0.90) | | 3.19 (1.23) |
| ARA (% of total fatty acids) | 12.18 (2.58) | 12.84 (2.52) | 12.00 (2.65) | 10.53 (2.26) | 11.04 (2.40) | 10.64 (2.36) | | 11.22 (2.56) |
| Global Proportion of Amerind ancestry | 0.06 | 0.06 | 0.12 | 0.33 | 0.39 | 0.41 | | 0.30 |
| Global Proportion of African ancestry | 0.19 | 0.41 | 0.23 | 0.09 | 0.16 | 0.04 | | 0.14 |
| Global Proportion of European ancestry | 0.75 | 0.53 | 0.65 | 0.58 | 0.45 | 0.55 | | 0.56 |
| rs174537 frequency* of effect allele T (versus G allele) | 0.28' | 0.27 | 0.40 | 0.56 | 0.59 | 0.59 | | 0.51 |

Phenotypic descriptive statistics are presented as percentages for dichotomous variables and mean (standard deviation) for continuous variables.
*For comparison, the rs174537 effect allele frequencies were 0.007, 0.328, and 0.858 in the 1000 Genomes AFR, EUR and AMR populations, respectively, where the allele frequency calculation was restricted to the cleaned set of samples that were included in the reference set for local ancestry analysis (see Supplementary Methods for details).

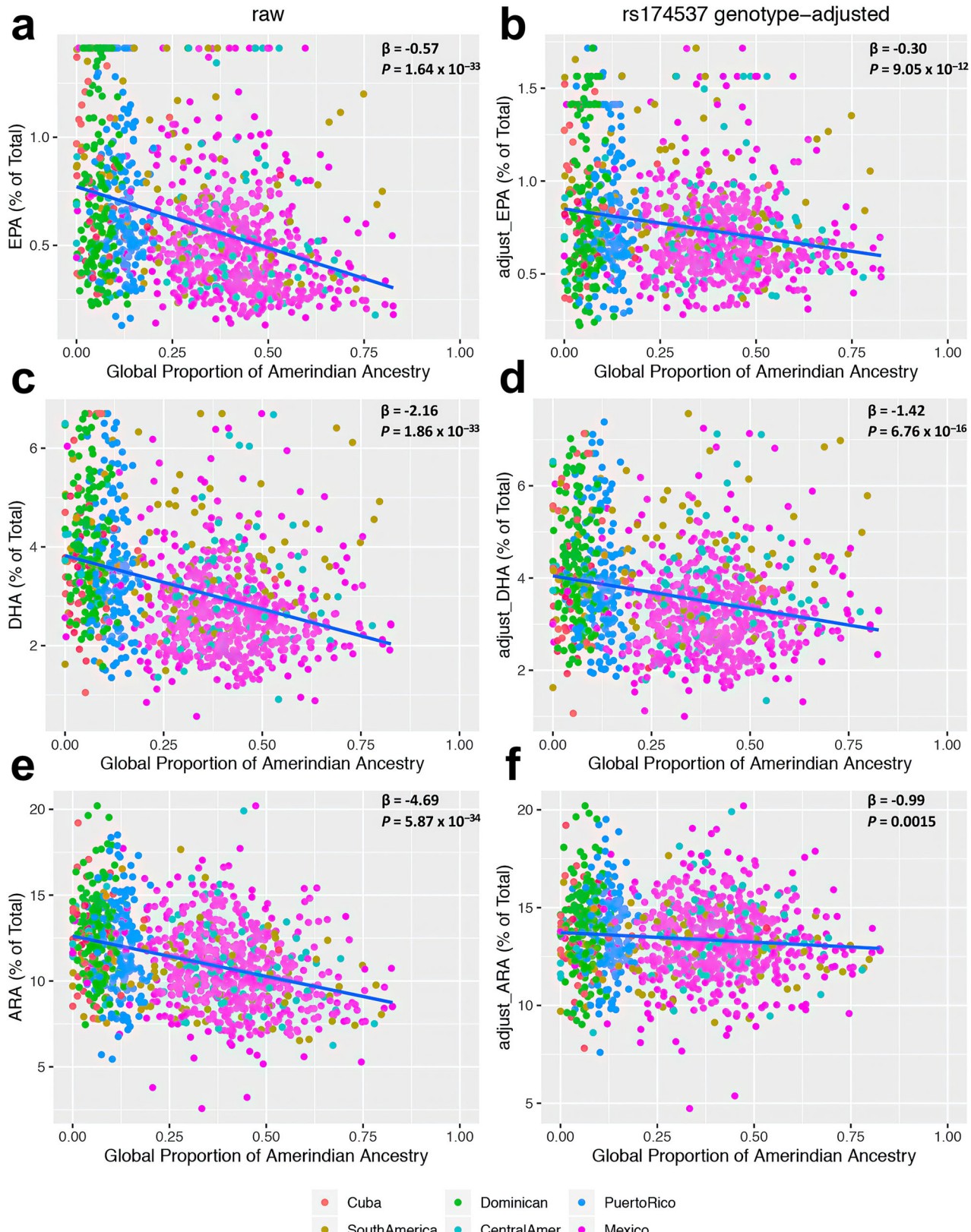

**Fig. 1 Relationship of LC-PUFA levels with global proportion of Amerind ancestry before and after adjustment for rs174537 genotype.** The regression effect estimates (β expressed as % of total fatty acids) and *P*-values are shown in the upper right corner of each panel. The relationship of LC-PUFA levels with Global Proportion of Amerind Ancestry as estimated from genome-wide SNP data is shown for **a** EPA—raw, **b** EPA—genotype-adjusted, **c** DHA—raw, **d** DHA—genotyped-adjusted, **e** ARA—raw, and **f** ARA—genotype-adjusted. Here, the rs174537 genotype-adjusted LC-PUFA levels were obtained as residuals after regression against rs174537 genotype and re-centered around the raw means. *P*-values are calculated using a two-sided *t*-test for the regression coefficient derived with *n* = 1102 biologically independent samples.

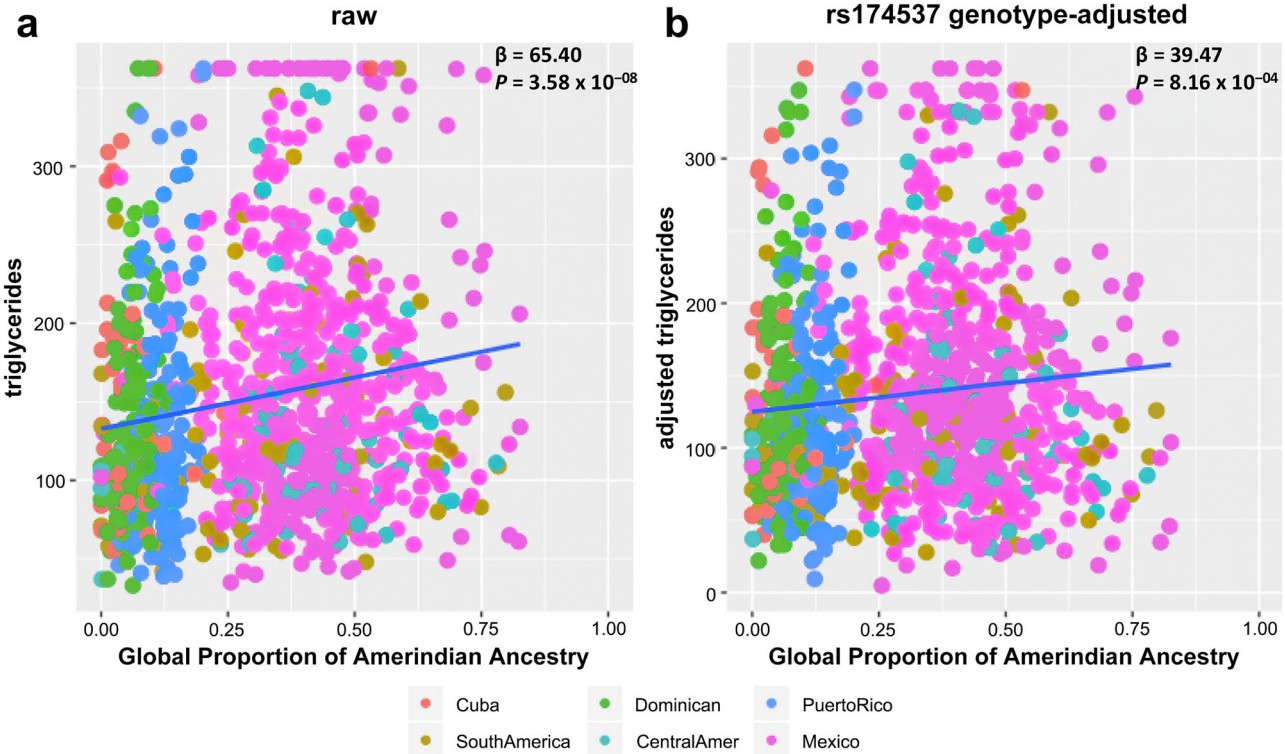

**Fig. 2 Relationship of triglycerides with global proportion of Amerind ancestry before and after adjustment for rs174537 genotype.** The regression effect estimates ($\beta$ in mg/dL) and *P*-values are shown in the upper right corner of each panel. The relationships are shown for each of **a** raw triglyceride levels, and **b** genotype-adjusted tryglyceride levels with Global Proportion of Amerind Ancestry. Here, rs174537 genotype-adjusted triglyceride levels were obtained as residuals from regression accounting for rs174537 genotype, and re-centered around the raw means. *P*-values are calculated using a two-sided *t*-test for the regression coefficient derived with *n* = 1101 biologically independent samples.

**Effects of rs174537 on inflammatory biomarkers, fasting lipids, and anthropometrics.** The effect of the *FADS* cluster SNP rs174537 on height, weight, body mass index (BMI), waist-hip ratio, s-ICAM, E-Selectin, and HDL-C was estimated in the MESA Hispanic participants. Initially, we examined unadjusted relationships which showed, for example, that mean E-selectin levels increased with the number of copies of the rs174537 effect allele T (Fig. 3b). In a model adjusted for age and sex, the rs174537 T allele was significantly associated with lower levels of HDL-C, higher waist-hip, lower height and weight, and higher levels of the inflammatory markers E-Selectin and s-ICAM (Table 2, Fig. 3d and Supplementary Table 3). In regression analysis with adjustment for principal components of ancestry, the rs174537 T allele remained significantly associated with higher TGs and lower height, while the associations with weight, waist-hip ratio, s-ICAM, E-Selectin, and HDL-C were no longer statistically significant (Supplementary Fig. 4 and Supplementary Table 3). In sensitivity analysis, we also examined the effect of rs174557 on the same set of phenotypes as examined for rs174537. Similar to the rs174537 T allele, the rs174557 A allele was significantly associated with lower levels of HDL-C, higher waist-hip, lower height and weight, and higher levels of the inflammatory markers s-ICAM (Supplementary Table 4 and Supplementary Fig. 4). In regression analysis with adjustment for principal components of ancestry, the rs174557 A allele remained significantly associated with higher TGs and lower height, while the associations with weight, waist-hip ratio, s-ICAM, E-Selectin, and HDL-C were no longer statistically significant (Supplementary Table 4).

**Replication in the AIR registry and HCHS/SOL cohort.** We conducted analyses in the AIR registry (*n* = 497) and HCHS/SOL

(*n* = 12,333) cohorts to examine the genotypic effect of rs174537 on multiple phenotypic traits including TGs and waist-to-hip ratio (Supplementary Tables 5, 6). In regression analyses adjusted for age and sex (and inclusion of random effects for household block and unit sharing in HCHS/SOL), the rs174537 T allele was significantly associated with TGs (AIR: $\beta = 10.4$ mg/dL, $P = 0.03$, HCHS/SOL: $\beta = 8.75$ mg/dL, $P = 5.84 \times 10^{-25}$ based on a two-sided *t*-test for the regression coefficient derived with *n* = 12,333) (Supplementary Table 7). The rs174537 T allele was also significantly associated with reduced height ($\beta = -1.33$, $P = 4.47 \times 10^{-56}$ calculated using a two-sided *t*-test for the regression coefficient derived with *n* = 12,333) and weight ($\beta = -1.25$, $P = 2.61 \times 10^{-08}$ calculated using a two-sided *t*-test for the regression coefficient derived with *n* = 12,333), and increased waist-to-hip ratio ($\beta = 0.003$; $P = 2.77 \times 10^{-05}$ calculated using a two-sided *t*-test for the regression coefficient derived with *n* = 12,333) in the HCHS/SOL cohort. The direction of effect was consistent, but not statistically significant, in the much smaller AIR cohort (Supplementary Table 7). The association of rs174537 with TGs remained statistically significant after adjustment for principal components of ancestry ($\beta = 4.05$ mg/dL, $P = 1.26 \times 10^{-05}$ calculated using a two-sided *t*-test for the regression coefficient derived with *n* = 497) and the effects were consistent across the HCHS/SOL study sites (Supplementary Table 8). We did not replicate these findings in the smaller AIR registry (Supplementary Table 7). S-ICAM and E-Selectin were not measured in either AIR or HCHS/SOL and thus could not be evaluated for replication of the findings from MESA.

## Discussion

While prior studies have identified genetic variants within the *FADS* locus with strong impact on fatty acid levels[48,49], prior

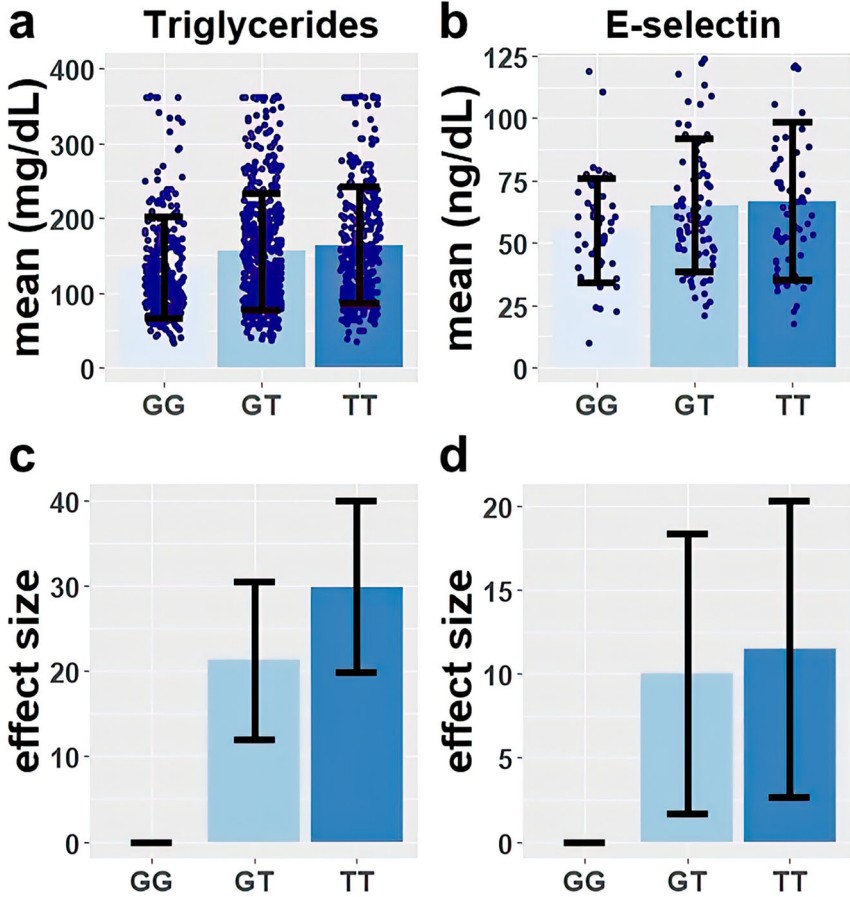

**Fig. 3 Genotypic effects of rs174537 on triglycerides and E-selectin.** The mean and standard deviation are shown for **a** triglycerides, and **b** E-selectin stratified by rs174537 genotypes. The estimated effect and standard error among participants carrying one or two copies of the ancestral allele T (compared to the reference of zero), after covariate adjustment for age and sex are shown for **c** triglycerides, and **d** E-selectin. Numbers of independent samples for the analyses presented are 1101 (GG: 293;GT: 483;TT: 325) for triglycerides and 183 (GG: 48; GT: 76; TT: 59) for E-selectin. Source data for the figure are provided in Supplementary Data 1.

literature has not examined directly the impact of population differences in allele frequencies on population-specific risk of fatty acid deficiency. In light of dramatic differences in genetic variation within the *FADS* locus across worldwide populations[56] and the marked changes in dietary n-6 and n-3 PUFA levels and ratios over the past 75 years, we carried out a study of to examine genomic proportion of AI ancestry as a predictor of n-3 and n-6 LC-PUFA levels and related cardiometabolic and inflammatory risk in the Hispanic participants from MESA. Our study first illustrates that certain Hispanic populations and particularly high AI-Ancestry populations have high frequencies of the ancestral allele at T at rs174537. Importantly, the frequency of the TT genotype associated with limited LC-PUFA biosynthesis ranges from <1% in African-Ancestry populations including African Americans to 40–55% in high AI-Ancestry Hispanics, and ~11% in European-Ancestry populations[65]. In light of high ancestral frequencies in certain Hispanic populations together with elevated dietary n-6 (LA) to n-3 (ALA) PUFAs ratios (>10:1) from the MWD entering the pathway, we postulated that these populations would be most likely to saturate their capacity to synthesize LC-PUFAs and particularly n-3 LC-PUFAs. Our statistical analyses demonstrated that global proportion of AI ancestry is predictive of reduced LC-PUFA phospholipid levels in the Hispanic population of the United States, accounting for ~12% of total variation in EPA, DHA, and ARA. Further, we showed that this relationship can be explained in large part by genetic variation within the *FADS* cluster. Given that many Hispanic

individuals will have reasonable knowledge of their AI ancestry, our work suggests a practical way to identify individuals likely to carry the homozygous TT genotypes, and for whom follow-up *FADS* genotyping assays may be warranted.

While both n-6 and n-3 LC-PUFAs are impacted, relatively high levels of ARA (~8.6% of total fatty acids) remain in circulating phospholipids in even the highest AI-Ancestry populations. In contrast, n-3 LC-PUFAs including EPA and DHA are reduced to the low (perhaps inadequate) levels of ~0.3% [EPA] and ~2% [DHA] of total fatty acids in circulating phospholipids in high AI-Ancestry individuals. It is not possible to say with certainty what levels of EPA and DHA or ratio of EPA + DHA/ARA would be inadequate (deficient) and have pathophysiologic impact, but these are certainly quantitatively very low concentrations and ratios of n-3 LC-PUFAs. It has been recognized that high levels of dietary LA relative to ALA from the modern Western diets (MWD) entering the LC-PUFA biosynthetic pathway are reciprocally related to levels of n-3 LC-PUFAs due to substate saturation of the enzymatic pathway[66,67]. Such a scenario was proposed by both Okuyama and colleagues and Lands and colleagues three decades ago to give rise to omega-3 deficiency syndrome and chronic pathophysiological events[20,47,68]. We propose that a limited LC-PUFA synthetic capacity in a greater proportion of AI-Ancestry Hispanics (due to the ancestral haplotype) in the context of excess dietary LA levels and high LA/ALA ratios renders inadequate n-3 LC-PUFAs more likely in this population.

**Table 2 Genotypic effects of rs174537 on fasting lipids, anthropometrics and inflammatory traits.**

| | | | Beta | P value |
|---|---|---|---|---|
| Fasting lipids | Triglycerides (mg/dL) | GT | 21.27 | 0.0001 |
| | | TT | 29.94 | $1.01 \times 10^{-06}$ |
| | HDL-C (mg/dL) | GT | −1.30 | 0.141 |
| | | TT | −2.48 | 0.010 |
| Anthropometrics | waist-hip ratio | GT | 0.006 | 0.152 |
| | | TT | 0.013 | $8.94 \times 10^{-03}$ |
| | Height (cm) | GT | −1.36 | 0.002 |
| | | TT | −3.46 | $6.59 \times 10^{-12}$ |
| | Weight (kg) | GT | −1.89 | 0.077 |
| | | TT | −3.12 | $7.48 \times 10^{-03}$ |
| | BMI (kg/m$^2$) | GT | −0.25 | 0.50 |
| | | TT | −0.002 | 0.99 |
| Inflammatory | s-ICAM (ng/mL) | GT | 30.64 | 0.002 |
| | | TT | 26.09 | 0.018 |
| | E-Selectin (ng/mL) | GT | 10.00 | 0.048 |
| | | TT | 11.50 | 0.032 |

Regression analysis results for the effect of rs174537 genotype on fasting lipids, anthropometrics and inflammatory with adjustment for age and sex. For the effect sizes, the effects of GT and TT are in reference to GG. The sample size is 1102 (GG: 293; GT: 484; TT: 325) for waist-hip ratio; height; weight and BMI, 1101 (GG: 293;GT: 483;TT: 325) for triglycerides and HDL-C, 439 (GG: 112; GT: 194; TT: 133) for s-ICAM and 183 (GG: 48; GT: 76; TT: 59) for E-Selectin. *P*-values are calculated using two-sided *t* test for the regression coefficient.

Our study also suggests that *FADS* variation has large effects on some critical cardiometabolic and inflammatory risk factors. Specifically, the proportion of AI ancestry was positively related to levels of circulating TGs and much of this effect was explained by variation in the *FADS* locus. While other studies have found associations between numerous genetic loci including *FADS* SNPs and circulating TGs[69–78], the high frequency of the ancestral *FADS* alleles (associated with elevated TGs) and their effect size in AI-Ancestry Hispanic populations that suggest that *FADS* variation is particularly relevant to TG levels in this population. The presence of the T allele at rs174537 had a large effect on circulating TG (GT vs GG: $\beta = 21.27$ mg/dL, $P = 0.0002$, TT vs GG: $\beta = 29.94$ mg/dL, $P = 1.01 \times 10^{-6}$) and this genotypic effect was replicated in both the AIR registry and HCHS/SOL cohort. Circulating TG are primarily synthesized in the liver and deficiencies of n-3 LC-PUFAs and imbalances of n-6 relative to n-3 PUFAs have been associated with elevated TGs and NAFLD[79]. Elevating n-3 LC-PUFA by diet or supplementation reduces TG by promoting hepatic fatty acid oxidation and reducing synthesis (via reducing *de novo* lipogenesis and decreasing fatty acid and adipokine release from adipocytes)[80–82]. These current data suggest that inadequate levels of n-3 LC-PUFAs in AI-Ancestry Hispanic populations may impact TG formation in the liver resulting in higher levels of circulating TG and potentially NAFLD.

Waist-to-hip ratio, used to describe the distribution of body fat, has been shown to be closely associated with hypertension, diabetes, dyslipidemia and cardiovascular disease[83]. A previous study examined genetic loci associated with BMI and waist-to-hip ratio and found nine BMI and seven central adiposity loci in Hispanic women[84]. To date, variation within *FADS* has not been associated with waist-to-hip ratio. While our study demonstrated that the ancestral rs174537 T allele was strongly associated with a higher waist-to-hip ratio and this risk factor was replicated in HCHS/SOL, the relationship was not statistically significant after adjusting for principal components of ancestry. Thus, waist-hip-ratio is an example of a trait for which the association with AI-Ancestry is not explained in large part by *FADS* variation.

The rs174537 allele T further demonstrated association with reduced height and weight in the large HCHS/SOL cohort (*n* =

12,333). Fumagalli and colleagues examined indigenous Greenland Inuit and found strong signals of natural selection within the *FADS* cluster[85]. The identified *FADS* variants were also strongly associated with anthropometric traits including body weight and height in the Inuit, and those associations were replicated in Europeans.

A wide variety of biomarkers of inflammation were measured in MESA, and there was a strong association between rs174537 and E-selectin which maintained suggestive evidence of association even after adjustment for population structure using genetic principal components of ancestry. E-Selectin (CD-62E) plays a pivotal role in the activation and adhesion of the migrating leukocytes to the endothelium[86]. These membrane bound adhesion molecules also undergo proteolytic cleavage that generate soluble forms that can be measured in the blood[87]. Serum levels of E-Selectin increase in many pathologies involving chronic inflammation including obesity[88], cardiovascular disease[89], bronchial asthma[90], and cancer[91,92].

Limitations of the study include a focus on primarily urban Hispanic American populations represented by the MESA cohort, potential confounding by diet and lifestyle habits across the six Hispanic subgroups in MESA, and systematic differences in PUFA levels across MESA study sites. To address the observable variation across Hispanic subgroups and study site, we included additional analyses stratified by these factors and demonstrated that our results were consistent across strata. Additionally, we used food frequency questionnaire data to confirm participants included in our analyses did not have self-reported use of fish oil supplements, and we performed analyses adjusted for self-reported fish intake in MESA. Still, we recognize there are inherent limitations with the quality of self-report-based measures of diet and supplement use. Further, we did not consider additional measures of dietary intake of n-3 and n-6 PUFAs in our regression analyses, in part because we determined that we did not have reliable measures available for these parameters in the MESA participants. Therefore, future studies should examine further the impact of dietary differences on the relationship between AI ancestry, *FADS* variation, and LC-PUFA levels.

Despite these limitations, our study reveals that *FADS* variation in AI-Ancestry Hispanic populations is inversely associated with dyslipidemia and inflammation, risk factors for a wide range of pathologies including cardiovascular and metabolic diseases. These associations are observed strongly in these Hispanic populations in part because of the high frequencies of ancestral *FADS* alleles. It may be that LC-PUFAs or their metabolites (eicosanoids, docosanoids, resolvins, protectins, etc.) are responsible for these genetic effects given the direct relationship between *FADS* variation and LC-PUFA levels. Alternatively, we have recently combined genetic and metabolomic analyses to identify the *FADS* locus as a central control point for biologically-active LC-PUFA-containing complex lipids that act as signaling molecules such as the endocannabinoid, 2-AG, and such endocannabinoids are known to impact anthropomorphic and other phenotypic characteristics[93].

Our results also suggest that targeting recommendations for n-3 and n-6 LC-PUFA intake/supplementation within AI-Ancestry Hispanic populations may be particularly effective. This premise is supported by the fact that numerous mechanistic studies directly link low levels of n-3 LC-PUFAs and high n-6 to n-3 ratios to elevated tissue and circulating TGs and NAFLD, and several recent reviews and meta-analyses suggest that n-3 LC-PUFA supplementation improves circulating and tissue levels of TG and NAFLD[94,95]. Prior research demonstrates that mean proportions of Amerind ancestry vary greatly by self-identified regions of origin among Hispanic Americans, with Mexican, Central American, and South American Hispanics showing the

greatest proportions, and individuals identifying as Cuban, Dominican, and Puerto Rican showing considerably lower proportions[61,96]. While a long term goal of applying precision nutrition may include genotyping of rs174537 (or related *FADS* region variants)_in routine health care screening, current health care practice does not provide adequate resources to genotype most individuals. Thus, a priori information predictive of ancestry such as country or origin or otherwise, may serve as a preliminary tool to prioritize those who are most likely to have low circulating and tissue levels of n-3 LC-PUFA and would benefit from additional screening either through genotyping or screening for n-3 LC-PUFA deficiency. Despite the current limitations of precision nutrition including inadequate genetic testing, the translational implications of this work are to point out that a large proportion of AI-Ancestry Hispanic populations have low (perhaps deficient) levels of n-3 LC-PUFAs and increased related risk factors. Thus, because of *FADS*-related deficiencies, these populations may be particularly responsive to diets or supplements enriched in n-3 LC-PUFAs.

## Methods

**Study participants**. MESA is a longitudinal cohort study of subclinical cardiovascular disease and risk factors that predict progression to clinically overt cardiovascular disease or progression of subclinical disease[57]. Between 2000 and 2002, MESA recruited 6814 men and women 45–84 years of age from Forsyth County, North Carolina; New York City; Baltimore; St. Paul, Minnesota; Chicago; and Los Angeles. Participants at baseline were 38% White, 28% African-American, 22% Hispanic, and 12% Asian (primarily Chinese) ancestry. This manuscript focuses on Hispanic American participants from MESA. Among the MESA Hispanic participants, self-reported birthplaces for parents' and grandparents' country/region of origin were used to assign country/region of origin to the following categories Central America, Cuba, the Dominican Republic, Mexico, Puerto Rico, and South American origin were assigned for the MESA Hispanic participants.

**Fatty acid measurements**. The fatty acids were measured by gas chromatography in EDTA plasma frozen at −70 °C[97].

Lipids were extracted from the plasma using a chloroform/methanol extraction method and the cholesterol esters, triglyceride, phospholipids, and free fatty acids are separated by thin-layer chromatography. The fatty acid methyl esters were obtained from the phospholipids and were detected by gas chromatography flame ionization. Individual fatty acids were expressed as a percent of total fatty acids. A total of 28 fatty acids were identified. Here, we focus on the following n-3 and n-6 fatty acids: eicosapentaenoic acid (EPA), docosapentaenoic acid (DPA), docosahexaenoic acid (DHA), and arachidonic acid (ARA).

**Additional phenotypes in MESA**. We considered additional phenotypes in analysis of the MESA data including lipids (HDL-C and triglycerides), anthropometric (height, weight, waist-hip ratio), and inflammatory markers (soluble E-Selectin and soluble ICAM-1). Details of measurement and treatment of outliers are provided in the Supplementary Methods and Supplementary Figs. 5, 6.

**Genotyping, genetic association, and ancestry analysis**. Participants in the MESA cohort who consented to genetic analyses and data sharing (dbGaP) were genotyped using the Affymetrix Human SNP Array 6.0 (GWAS array) as part of the NHLBI SHARe (SNP Health Association Resource) project. Genotype quality control for these data included filter on SNP level call rate <95%, individual level call rate <95%, heterozygosity >53%[98]. The cleaned genotypic data was deposited with MESA phenotypic data into dbGaP (study accession phs000209.v13.p3); 8224 consenting individuals (2685 White, 2588 non-Hispanic African-American, 2174 Hispanic, 777 Chinese) were included, with 897,981 SNPs passing study specific quality control (QC). SNP coverage from the original GWAS SNP genotyping array was increased through imputation using the 1,000 Genomes Phase 3 integrated variant set completed using the Michigan Imputation Server (https://imputationserver.sph.umich.edu).

Prior studies have highlighted multiple different *FADS* variants for their role in regulation of fatty acid synthesis, including rs174537[48] and rs174557;[64] however, the relevant variants at the primary signal within the *FADS* region exhibit extended linkage disequilibrium across the region[99]. Therefore, we focused our genetic analyses primarily on the variant rs174537, with additional sensitivity analyses using similar models for the variant rs174557. Imputed genotype data were used for genetic association analysis of the rs174537 and rs174557 SNPs (for which the imputation R-squared in MESA Hispanics were both 0.99). Our statistical analyses used genotype dosage information from imputation, except where noted otherwise. For analyses that required us to stratify by genotype, including those presented in

Table 2 and Fig. 3, we used the estimated most likely genotype from imputation. Principal components of ancestry were computed using genome-wide genotype data[98]. Global proportions of Amerind ancestry were estimated in MESA participants by leveraging reference samples from the 1000 Genomes[100] and the Human Genome Diversity Project (HGDP)[101,102]. Local ancestry for each individual was defined as the genetic ancestry at the position of *FADS* SNP rs174537, where each individual can have 0, 1, or 2 copies of an allele derived from each of the three possible ancestral populations (European, African and Amerind). Local ancestry, was estimated using the RFMix package[103]. Details are provided in the Supplementary Methods.

**Regression modeling of n-3 and n-6 PUFAs**. As we observed a strong effect of study site in regression analysis of all LC-PUFAs (Supplementary Table 9), we performed regression analyses stratified by study site and combined by inverse-variance weighted meta-analysis. In order to examine the effect of global Amerind ancestry on the levels of n-3 and n-6 PUFAs in the MESA Hispanic participants, we carried out linear regression analyses using three different models for each of the PUFA levels as follows:

(1) PUFA ~ age + sex + fish intake + global proportion of Amerind ancestry,
(2) PUFA ~ age + sex + fish intake + global proportion of Amerind ancestry + rs174537 genotype, and
(3) PUFA ~ age + sex + fish intake + global proportion of Amerind ancestry + rs174537 genotype + *FADS* region local proportion of Amerind ancestry.

**Regression modeling for genotypic effects of *FADS* cluster SNP rs174537 on proximal traits**. To examine the effect of *FADS* SNP rs174537 on lipids (HDL-C and triglycerides), anthropometric (height, weight, and waist-to-hip ratio), and inflammatory markers (s-ICAM and E-Selectin) in MESA Hispanic participants, we performed linear regression analysis with covariate adjustment for (1) age and sex, and (2) age, sex and the first four principal components of ancestry.

**Replication analysis in the AIR registry and HCHS/SOL cohort**. We conducted follow-up regression analyses to examine the association of rs174537 with phenotypic traits in both the AIR registry and the HCHS/SOL cohort. The variant rs174537 was genotyped directly in both AIR and HCHS//SOL. Details are provided in the Supplementary Methods.

**Statistics and reproducibility**. All of our statistical analyses were carried out on biologically independent samples from MESA ($n = 1102$), AIR ($n = 497$) and HCHS/SOL ($n = 12,333$). Analyses in MESA were carried out for an unrelated subset of participants constructed by retaining at most one individual from each group of first-degree relatives. We did not perform relationship inference and removal of first-degree relatives in AIR, as there were no genome-wide data available to infer relationship among individuals. In HCHS/SOL, all individuals (both related and unrelated) were included in analyses, as we accounted for their relationships using linear mixed models. Regression analyses presented throughout the manuscript included adjustments for relevant covariates as stated for each model presented in the text.

**Ethical review**. All MESA participants provided written informed consent for participation at their respective MESA study sites, and the MESA study was also reviewed and approved by the Institutional Review Boards (IRBs) at each of the participating study sites. The current investigation including activities for analysis of LC-PUFA levels in MESA was reviewed and approved by the Institutional Review Board (IRB) at the University of Virginia. The AIR registry was approved by the IRB at the University of Arizona and all subjects gave written informed consent before their participation. The HCHS/SOL was approved by the IRBs at all participating institutions including the Albert Einstein College of Medicine, and all participants gave written informed consent.

**Reporting summary**. Further information on research design is available in the Nature Research Reporting Summary linked to this article.

## Data availability
Genome-wide genotype data for the Multi-Ethnic Study of Atherosclerosis (MESA)[57,61,98] and the Hispanic Community Health Study/Study of Latinos (HCHS/SOL)[60,96,104] are available by application through dbGaP. The dbGaP accession numbers are: MESA phs000209 and HCHS/SOL phs000810. Source data underlying figures are presented in Supplementary Data 1–3. All other data are available from the corresponding author (or other sources, as applicable) on reasonable request.

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

## Acknowledgements

This work was supported by NCCIH R01 AT008621 and USDA ARZT-1361680-H23-157. *The Multi-Ethnic Study of Atherosclerosis:* MESA and the MESA SHARe project are conducted and supported by the National Heart, Lung, and Blood Institute (NHLBI) in collaboration with MESA investigators. Support for MESA is provided by contracts HHSN268201500003I, N01-HC-95159, N01-HC-95160, N01-HC-95161, N01-HC-95162, N01-HC-95163, N01-HC-95164, N01-HC-95165, N01-HC-95166, N01-HC-95167, N01-HC-95168, N01-HC-95169, UL1-TR-000040, UL1-TR-001079, UL1-TR-001420, UL1-TR-001881, and DK063491. Funding for SHARe genotyping was provided by NHLBI Contract N02-HL-64278. Genotyping was performed at Affymetrix (Santa Clara, California, USA) and the Broad Institute of Harvard and MIT (Boston, Massachusetts, USA) using the Affymetrix Genome-Wide Human SNP Array 6.0. *The Hispanic Community Health Study/Study of Latinos:* HCHS/SOL is a collaborative study supported by contracts from the National Heart, Lung, and Blood Institute (NHLBI) to the University of North Carolina (HHSN268201300001I/N01-HC-65233), University of Miami (HHSN268201300004I/N01-HC-65234), Albert Einstein College of Medicine (HHSN268201300002I/N01-HC-65235), University of Illinois at Chicago (HHSN268201300003I/N01-HC-65236 Northwestern Univ), and San Diego State University (HHSN268201300005I/N01-HC-65237). The following Institutes/Centers/Offices have contributed to the HCHS/SOL through a transfer of funds to the NHLBI: National Institute on Minority Health and Health Disparities, National Institute on Deafness and Other Communication Disorders, National Institute of Dental and Craniofacial

Research, National Institute of Diabetes and Digestive and Kidney Diseases (NIDDK), National Institute of Neurological Disorders and Stroke, and NIH Institution-Office of Dietary Supplements. The authors thank the staff and participants of HCHS/SOL for their important contributions. A complete list of staff and investigators has been provided by Sorlie P., et al. in *Ann. Epidemiol.* 2010 Aug;20: 642–649 and is also available on the study website http://www.cscc.unc.edu/hchs/. Other funding sources for this study include R01HL060712, R01HL140976, and R01HL136266 from the NHLBI; and R01DK119268, R01DK120870 and the New York Regional Center for Diabetes Translation Research (P30 DK111022) from the NIDDK. The Genetic Analysis Center at the University of Washington was supported by NHLBI and NIDCR contracts (HHSN268201300005C AM03 and MOD03). Genotyping efforts were supported by NHLBI HSN 26220/20054C, NCATS CTSI grant UL1TR000123, and NIDDK Diabetes Research Center (DRC) grant DK063491. *The Arizona Insulin Resistance Registry:* The AIR registry was supported by Health Research Alliance Arizona and the Center for Metabolic Biology at Arizona State University. Data management support was provided by a grant (UL1 RR024150) from the Mayo Clinic to utilize Research Electronic Data Capture (REDCap).

## Author contributions

Y.I.C., A.M.F., S.A.B., L.M.J., and C.T. generated the data. C.Y., B.H., J.C.C., Q.Q., and A.M. analyzed the data. C.Y., B.H., J.C., T.D.O., L.M.R., A.C.W., M.S., M.Y.T., R.N.L., D.K.C., L.M.J., Q.Q., I.R., S.S.R., R.A.M., F.H.C., and A.M. contributed to interpretation of the data. F.H.C., A.M., R.A.M., and S.S.R. conceptualized and designed the study. Y.I.C., L.M.S., M.Y.T., R.C.K., M.L.D., L.J.M., D.K.C., and S.S.R. provided critical oversight to data collection and study coordination. C.Y., B.H., A.M., and F.H.C. wrote the manuscript. All authors contributed to critical editing of the manuscript.

## Competing interests

Floyd H. Chilton is a co-founder of Tyrian Omega Inc. All other authors declare no competing interests.
