## [Peer Review File · Communications Biology]

Reviewers' comments:

Reviewer #1 (Remarks to the Author):

In this manuscript, Yang and colleagues present a comprehensive evaluation regarding the impact of Amerind ancestry on the effect of genetic variants in the known FADS gene cluster on deficiency of long chain polyunsaturated fatty acids (LC-PUFAs), related lipids trait, cardiometabolic and inflammatory risk in Hispanic populations. If I understand correctly, the logic follows like below: FADS variants associated with more limited capacity to synthesize LC-PUFAs have been discovered. These variants are nearly fixed in Native American and Greenland Inuit populations (presumably with the risk allele reaching frequency close to 1) and have much allele frequency in Amerind (AI) Ancestry Hispanic populations. Given the above two pieces of information, the authors set out to test their hypothesis that Hispanic individuals carrying FADS risk alleles and with low levels of LC-PUFAs are more likely to be of Amerind ancestry.

There are many merits of the study. For example, the scope is rather broad in that the authors not only carefully assessed the effect on LC-PUFAs but also on triglycerides, Inflammatory Biomarkers, Fasting Lipids and Anthropometrics traits. For another example, the authors fit three models trying to carefully discern allelic effect (due to genotypes of genetic variants in FADS), local ancestry effect and global ancestry effect. For yet another example, the authors also evaluated all other variants in the FADS to identify variants other than the top one rs174537. Finally, the authors validated their findings in two additional cohorts: the AIR registry and the HCHS/SOL cohort.

The manuscript was clearly written, technically sound and with sufficient details. I was able to find almost every important technical detail either in the main or in the supplementary materials.

The only major comment I have is regarding the reference panel for local ancestry inference (maybe also for global, which the authors didn't seem to explicitly specify the reference panel). I clearly see that the authors made efforts to accommodate Native American ancestry by including PEL in the reference. I wonder whether including larger number and more diverse Amerind ancestry individuals from projects such as PAGE or HGDP would help making better inference.

I also have several minor questions/comments.

- (1) There is a very drastic allele frequency difference at rs174537 (<https://bravo.sph.umich.edu/freeze8/hg38/variant/snv/11-61785208-G-T#>) : T allele rather low frequency in AFR (only ~2.5% frequency) while much higher or even major in other populations (16-59% frequency). Any evidence of selection?
- (2) How accurate are the local ancestry estimates at rs174537?
- (3) Have the authors evaluated any interaction effect among rs174537, local or global ancestry?

Reviewer #2 (Remarks to the Author):

This paper describes an association of genetic variability in the FADS region with PUFA levels and some disease phenotypes in a set of Hispanic population groups with different origins. Their main result, as stated in the abstract, is "We demonstrate strong negative associations between AI genetic ancestry and LC-PUFA levels. The FADS rs174537 single nucleotide polymorphism (SNP) accounted for much of the AI ancestry effect on PUFAs, especially for low levels of n-3 LC-PUFAs. Rs174537 was also strongly associated with several metabolic, inflammatory and anthropomorphic traits including circulating triglycerides (TGs) and E-selectin in MESA Hispanics."

Unfortunately, this study is poorly designed and executed and the results very predictable and not novel. First, the study is based on a fairly small sample size, including subpopulations with Hispanics

from a set of American populations, resulting in limited statistical power. Second, with regard to the FADS region the authors apparently have not followed the scientific literature and failed to notice that the FADS region has been studied in detail and the functional SNP regulating FADS gene expression was identified already in 2017 (<https://pubmed.ncbi.nlm.nih.gov/27932482/>). Studies aimed to address the importance of the FADS region should be based on the SNP actually involved, rather than genotype information from imputed data. The use of an imputed genotype in the analysis, rather than the one actually shown to regulate FADS expression levels, not only reduces the statistical power but could lead to erroneous results in terms of understanding the importance of the genetic variation for the traits measured. In terms of the haplotype nomenclature, the authors also fail to cite the correct reference and those introducing the nomenclature of the ancestral and derived haplotypes (<https://pubmed.ncbi.nlm.nih.gov/22503634/>), but rather pick a review of the literature.

Their main conclusion in the abstract is that most of the "ancestry effect" on PUFA levels is accounted for by the SNP in the FADS region. In reality, after correcting for the FADS genotype effect very little variation remains to be explained by the "ancestry effect". I believe this study design is incorrect. First, the authors should have used an updated genetic analysis, focusing on the actual functional SNP rather than an imputed genotype. Second, the PUFA levels should have been corrected for genotype frequency, and then any residual effects examined. As shown by the authors the so called "ancestry effect" is merely a reflection of the frequency of the different haplotypes (A and D) in the different subpopulations. There might well be other covariates affecting the PUFA levels in addition to the genotype, but in order to have the necessary power to identify these covariates, given the very dominant effect of the FADS haplotype, much larger epidemiological studies have to be performed. Finally, the authors state "Our study demonstrates that Amerind ancestry provides a useful and readily available tool to identify individuals most likely to have FADS related LC-PUFA deficiencies and associated cardiovascular risk." While there might be differences among the population groups in the frequency of the FADS haplotypes, the authors do not explain how this information could actually be used in preventive medicine. In reality, any intervention in diet or lifestyle will have to be based on a "personalized medicine" level, and genetic analysis interrogating the correct genetic variant in each individual.

Reviewer #3 (Remarks to the Author):

- General

- o The authors investigate the association between Amerind ancestry and FADS rs174537 genotype with LC-PUFA levels as well as cardiometabolic characteristics in self-identified Hispanics of the MESA cohort. The manuscript is well written and the findings address an important gap that would benefit the field. Mostly only minor revisions are necessary.

- Major comments

- o A statement in the discussion regarding the knowledge of one's ancestry requires further explanation to avoid misinterpretation from future readers or should be removed (lines 339-342; also lines 116-118 in the abstract). Is there any literature that suggests self-identified Hispanics have reasonable knowledge of their proportion ancestry (beyond what may be guessed from country/region of origin)? If not, are the authors suggesting with this statement that region of origin could be used as a screening tool to select candidates for follow-up FADS genotyping (e.g., Mexicans but not Puerto Ricans should be screened)? Without further explanation, the idea of self-identified ancestry as a translational tool for FADS function sounds far-fetched. In contrast, the genotype of rs174537 alone would be cheap and substantially more accurate.

- Minor comments

- o I would suggest that the authors add body mass index. This is a more interpretable cardiometabolic

factor compared to individual height and weight measurements.

o Figure 3 and S2 are hard to interpret: please provide the sample size across genotype, provide error bars for the upper panels that show mean levels by genotype (or explain why error bars are not needed), and describe what the error bars indicate for the lower panels that show the effect size (standard deviation?). For the effect sizes, it looks like the effects of TT are in reference to GG (as opposed to GT), but this needs to be explicitly stated.

o The reference to "(Figure 2 -left)" at lines 279-280 is in the wrong place, I believe. It should be earlier when the unadjusted regression is described.

o Table 2: the reference group for the genotypic effects (FADS GG genotype?) should be explicitly clear

o Self-reported fish intake was measured in MESA and appropriately adjusted for by the investigators. Is the use of fish oil or other supplements that impact PUFA levels and associated metabolites available in MESA. If not, this should also be listed as a limitation. This is partially covered under the "diet and lifestyle" limitations that the authors state, but should be more specifically mentioned. Based on the histograms in the supplement that show a small subset of participants with very high PUFA levels, it seems likely that supplements may be playing a role here.

Re: COMMSBIO-20-3297-T “Amerind ancestry predicts the impact of *FADS* genetic variation on omega-3 PUFA deficiency, cardiometabolic and inflammatory risk in Hispanic populations”

April 15, 2021

Dear Reviewers,

We are grateful for your careful and thoughtful review of our manuscript. We have updated our manuscript in response to the major suggestions and comments from the first round of review.

Specifically, in response to a suggestion from Reviewer #1, we have combined the 1000 Genomes populations with the Human Genome Diversity Project (HGDP) whole genome sequence data (just published and released in 2020) to construct an expanded reference panel for global and local ancestry analysis. Based on the new reference panel, we obtain more statistically significant results for the relationship between global proportion of Amerind ancestry and LC-PUFA levels as well as triglycerides.

In response to suggestions from Reviewers #2 and #3, we have expanded the Discussion to clarify the relationship between country of origin and Amerind ancestry, as well as the role of Amerind ancestry as a tool to identify individuals likely to have *FADS*-related LC-PUFA deficiencies and related risk factors.

Based on the point from Reviewer #3 regarding the suggestion that other *FADS* variants should be examined, we have repeated the major analyses of our paper (previously focused on *FADS* variant rs174537) using the variant rs174557. These additional analyses demonstrate that we reach similar conclusions regarding the important role of *FADS* variants in the relationship between Amerind ancestry and LC-PUFA levels, even when focusing on a different variant that exhibits partial but not complete linkage disequilibrium with the variant examined previously.

We have included detailed replies to each of the reviewers' comments below. We hope you agree that we have addressed fully the concerns and suggestions raised in the prior round of review.

Your sincerely,

Ani Manichaikul

Reviewers' comments

Reviewer #1

Major comment:

In this manuscript, Yang and colleagues present a comprehensive evaluation regarding the impact of Amerind ancestry on the effect of genetic variants in the known FADS gene cluster on deficiency of long chain polyunsaturated fatty acids (LC-PUFAs), related lipids trait, cardiometabolic and inflammatory risk in Hispanic populations. If I understand correctly, the logic follows like below: FADS variants associated with more limited capacity to synthesize LC-PUFAs have been discovered. These variants are nearly fixed in Native American and Greenland Inuit populations (presumably with the risk allele reaching frequency close to 1) and have much allele frequency in Amerind (AI) Ancestry Hispanic populations. Given the above two pieces of information, the authors set out to test their hypothesis that Hispanic individuals carrying FADS risk alleles and with low levels of LC-PUFAs are more likely to be of Amerind ancestry.

There are many merits of the study. For example, the scope is rather broad in that the authors not only carefully assessed the effect on LC-PUFAs but also on triglycerides, Inflammatory Biomarkers, Fasting Lipids and Anthropometrics traits. For another example, the authors fit three models trying to carefully discern allelic effect (due to genotypes of genetic variants in FADS), local ancestry effect and global ancestry effect. For yet another example, the authors also evaluated all other variants in the FADS to identify variants other than the top one rs174537. Finally, the authors validated their findings in two additional cohorts: the AIR registry and the HCHS/SOL cohort.

The manuscript was clearly written, technically sound and with sufficient details. I was able to find almost every important technical detail either in the main or in the supplementary materials.

Response: We thank the reviewer for the careful reading of the manuscript and the positive overall assessment and constructive feedback. We have incorporated the reviewer's specific suggestions as detailed below.

1. The only major comment I have is regarding the reference panel for local ancestry inference (maybe also for global, which the authors didn't seem to explicitly specify the reference panel). I clearly see that the authors made efforts to accommodate Native American ancestry by including PEL in the reference. I wonder whether including larger number and more diverse Amerind ancestry individuals from projects such as PAGE or HGDP would help making better inference.

Response: We agree with the reviewer and have combined the 1000 Genomes populations and the Human Genome Diversity Project (HGDP) whole genome sequence data (just published and released in 2020) to construct an appropriate reference panel for global and local ancestry analysis. Based on the new reference panel, we obtain more statistically significant results for the relationship between global proportion of Amerind ancestry and LC-PUFA levels (updated Figure 1) as well as triglycerides (updated Figure 2).

Minor comments:

1. There is a very drastic allele frequency difference at rs174537

(<https://bravo.sph.umich.edu/freeze8/hg38/variant/snv/11-61785208-G-T#>) : T allele rather low frequency in AFR (only ~2.5% frequency) while much higher or even major in other populations (16-59% frequency). Any evidence of selection?

Response: We agree the allele frequency differences for the rs174537 variant are striking. Among others, Harris *et al.* (PMID: 30942856) has demonstrated that Amerind ancestry is nearly fixed for the ancestral haplogroup of *FADS*, and further confirm a positive selection signal in Native American populations. We have now updated the Introduction (p. 8, lines 181-185) to provide more information on selection within the *FADS* region:

'FADS variants associated with more limited capacity to synthesize LC-PUFAs (termed "ancestral" haplotype) are nearly fixed in Native American and Greenland Inuit populations and found at high frequencies in Amerind (AI) Ancestry Hispanic populations.⁵⁶ These distinct patterns of haplotypes have resulted in part from positive selection for the ancestral haplotype among Indigenous American populations.⁵⁶

2. How accurate are the local ancestry estimates at rs174537?

Response: Unfortunately, we do not know the "true" local ancestry values to estimate accuracy in our own study. In simulation studies from the of RFMix paper (PMID: 23910464), they demonstrate that RFMix can achieve ~93-95% diploid accuracy in identification of local continental ancestry for Hispanic/Latino samples.

3. Have the authors evaluated any interaction effect among rs174537, local or global ancestry?

Response: We have conducted interaction analysis in a linear regression framework to examine global and local ancestry as potential modifiers of the effect of rs174537 on circulating fatty acid levels. The analyses were carried out stratified by study site as well as combined by meta-analysis. We did not find evidence of interaction for rs174537 dosage with either local or global ancestry (Results p. 11, lines 254-256): *"In additional analysis examining global and local ancestry as potential modifiers of the effect of rs174537 on circulating fatty acid levels, we did not observe statistically significant evidence of interaction"* and new **Table S2**.

Reviewer #2

Major comment:

This paper describes an association of genetic variability in the FADS region with PUFA levels and some disease phenotypes in a set of Hispanic population groups with different origins. Their main result, as stated in the abstract, is “We demonstrate strong negative associations between AI genetic ancestry and LC-PUFA levels. The FADS rs174537 single nucleotide polymorphism (SNP) accounted for much of the AI ancestry effect on PUFAs, especially for low levels of n-3 LC-PUFAs. Rs174537 was also strongly associated with several metabolic, inflammatory and anthropomorphic traits including circulating triglycerides (TGs) and E-selectin in MESA Hispanics.”

Unfortunately, this study is poorly designed and executed and the results very predictable and not novel.

Response: We respectfully disagree with the reviewer on this point. This manuscript first demonstrates that *FADS* variations impact the capacity of individuals to synthesize biologically-active n-3 LC-PUFAs and their metabolites from dietary PUFAs. This was not surprising. However, the manuscript also provides critical data needed to fill unanswered gaps in our knowledge in two areas. First, it provides evidence that the previously proposed ‘Omega-3 Deficiency Syndrome’ exists in a high proportion of individuals in Amerind Ancestry and much of this is driven by the frequency of the ancestral *FADS* haplotype in this Hispanic populations. Perhaps more unexpectedly, ancestral *FADS* variation (and low n-3 LC-PUFA levels) were shown to be associated with anthropometric characteristics, inflammatory biomarkers, and increased cardiometabolic risk factors in these Amerind ancestry populations. Taken together, data in this manuscript emphasizes the need for population/individual-based approaches to dietary recommendations and the future design of supplementation trials, or to achieve a healthy balance of n-3 and n-6 PUFAs in order to reduce inflammatory and metabolic disease risk factors. Collectively, we believe this is vital information.

We do agree with the reviewer that we could have done a better job of emphasizing the novelty of our manuscript is not in the discovery of specific *FADS* variants in relation to LC-PUFA levels, but in the concept of relating these major variants to Amerind ancestry and how they impact anthropomorphic, inflammatory and metabolic disease risk factors. We have now clarified these points in the Introduction (p. 8, lines 186-189): “*While the role of FADS variation in modulating circulating fatty acid levels has been documented previously^{48,49}, prior studies have not examined the impact that population differences in FADS allele frequencies have in downstream population-specific risk of fatty acid deficiency*” and Discussion (p. 13, lines 323-325): “*While prior studies have identified genetic variants within the FADS locus with strong impact on fatty acid levels^{48,49}, prior literature has not examined directly the impact of population differences in allele frequencies on population-specific risk of fatty acid deficiency.*”

1. First, the study is based on a fairly small sample size, including subpopulations with Hispanics from a set of American populations, resulting in limited statistical power.

Response: While a larger sample size can always provide advantages over the available sample size, we emphasize that the sample size examined in our current study is more than adequate to make clear the major point that Amerind ancestry is a major predictor of carrying *FADS* genotypes predisposing certain individuals to LC-PUFA deficiencies. While many prior studies of *FADS* haplotypes have focused on populations with smaller degrees of admixture, the

admixed nature of our Hispanic study sample makes it ideal in addressing our scientific question of interest in the current work.

2. Second, with regard to the FADS region the authors apparently have not followed the scientific literature and failed to notice that the FADS region has been studied in detail and the functional SNP regulating FADS gene expression was identified already in 2017 (<https://pubmed.ncbi.nlm.nih.gov/27932482/>). Studies aimed to address the importance of the FADS region should be based on the SNP actually involved, rather than genotype information from imputed data. The use of an imputed genotype in the analysis, rather than the one actually shown to regulate FADS expression levels, not only reduces the statistical power but could lead to erroneous results in terms of understanding the importance of the genetic variation for the traits measured. In terms of the haplotype nomenclature, the authors also fail to cite the correct reference and those introducing the nomenclature of the ancestral and derived haplotypes (<https://pubmed.ncbi.nlm.nih.gov/22503634/>), but rather pick a review of the literature.

Response: We are familiar with the paper referenced by the Reviewer (PMID: 27932482) as we also have recently published a paper on the tissue-specific impact of *FADS* variants on *FADS1* and *FADS2* expression (PMID: 29590160). Due to the long range haplotype structure in the *FADS* region, the issue of identifying a single functional *FADS* variant remains elusive. The main variant indicated by the reviewer is rs174557, and we found rs174548.

We examined the linkage disequilibrium (LD) between rs174557 and rs174537 (the primary variant used for analyses in our current manuscript). These variants are indeed in LD, but the LD is partial rather than complete. Specifically, the LD R-squared is 0.90, 0.83, 0.81, 0.93 in the self-reported MESA Hispanic, African American, European and Chinese, respectively. We note that the imputation R-squared for rs174537 and rs184557 were both 0.99 in MESA Hispanics.

However, to completely address the Reviewer's points regarding rs174557, we have repeated the analysis of our main Figures 1-2 and Table 2 using the rs174557 variant. The results of these analyses are now shown in the new **Supplementary Figures S1-S2**, and **Table S4**.

Updated Figure S1: Relationship of LC-PUFA levels with Global Proportion of Amerind Ancestry before and after adjustment for rs174557 genotype.

Updated Figure S2: Relationship of triglycerides with Global Proportion of Amerind Ancestry before and after adjustment for rs174557 genotype.

Similar results were obtained when these analyses are performed for rs174537 versus rs174557; see Results (p. 11, lines 257-264):

*“As other studies have suggested different specific variants as potentially functional within the FADS region, we further repeated the analysis presented in **Figure 1** through sensitivity analysis focused on the FADS region variant rs174557 (**Figure S1**), a common variant that diminishes binding of PATZ1, a transcription factor conferring allele-specific downregulation of FADS1.⁶⁴ After adjusting for rs174557 genotype, we observed association between global proportion of AI ancestry and LC-PUFA levels (EPA: $\beta = -0.30$, $P = 2.19 \times 10^{-11}$; DHA: $\beta = -1.39$, $P = 2.54 \times 10^{-15}$; ARA: $\beta = -1.02$, $P = 0.0012$) similar to that seen after adjusting for rs174537.”*

and (pp. 12-13, lines 296-303):

*“In sensitivity analysis, we also examined the effect of rs174557 on the same set of phenotypes as examined for rs174537. Similar to the rs174537 T allele, the rs174557 A allele was significantly associated with lower levels of HDL-C, higher waist-hip, lower height and weight, and higher levels of the inflammatory markers s-ICAM (**Table S4 and Figure S4**). In regression analysis with adjustment for principal components of ancestry, the rs174557 A allele remained significantly associated with higher TGs and lower height, while the associations with weight, waist-hip ratio, s-ICAM, e-Selectin and HDL-C were no longer statistically significant (**Table S4**).”*

3. Their main conclusion in the abstract is that most of the “ancestry effect” on PUFA levels is accounted for by the SNP in the FADS region. In reality, after correcting for the FADS genotype effect very little variation remains to be explained by the “ancestry effect” . I believe this study design is incorrect. First, the authors should have used an updated genetic analysis, focusing on the actual functional SNP rather than an imputed genotype. Second, the PUFA levels should have been corrected for genotype frequency,

and then any residual effects examined. As shown by the authors the so called “ancestry effect” is merely a reflection of the frequency of the different haplotypes (A and D) in the different subpopulations. There might well be other covariates affecting the PUFA levels in addition to the genotype, but in order to have the necessary power to identify these covariates, given the very dominant effect of the FADS haplotype, much larger epidemiological studies have to be performed. Finally, the authors state “Our study demonstrates that Amerind ancestry provides a useful and readily available tool to identify individuals most likely to have FADS related LC-PUFA deficiencies and associated cardiovascular risk.” While there might be differences among the population groups in the frequency of the FADS haplotypes, the authors do not explain how this information could actually be used in preventive medicine. In reality, any intervention in diet or lifestyle will have to be based on a “personalized medicine” level, and genetic analysis interrogating the correct genetic variant in each individual.

Response: We agree with the reviewer that there is very little ancestry effect remaining after accounting for the *FADS* variant, and that is the point that we make in **Figure 1**. The focus of our study is to demonstrate that Amerind ancestry is a strong predictor of *FADS* genotype, which can provide a useful approach to identify individuals at greater risk of carrying genotypes that lead to LC-PUFA deficiencies as well as inflammatory and cardiometabolic risk. Until preventive medicine has the capacity to interrogate specific genetic variants in individuals especially in highly underserved Amerind populations, such as Mexican Americans, we believe it is vital to understand which populations are most likely to be impacted by omega-3 deficiency and related risk factors. We have significantly expanded the Discussion (p. 18, lines 430-445) to clarify the proposed approach of how Amerind ancestry could be used as a tool to identify individuals most likely to have *FADS*-related LC-PUFA deficiencies.

Reviewer #3

The authors investigate the association between Amerind ancestry and FADS rs174537 genotype with LC-PUFA levels as well as cardiometabolic characteristics in self-identified Hispanics of the MESA cohort. The manuscript is well written and the findings address an important gap that would benefit the field. Mostly only minor revisions are necessary.

Major comment:

1. A statement in the discussion regarding the knowledge of one's ancestry requires further explanation to avoid misinterpretation from future readers or should be removed (lines 339-342; also lines 116-118 in the abstract). Is there any literature that suggests self-identified Hispanics have reasonable knowledge of their proportion ancestry (beyond what may be guessed from country/region of origin)? If not, are the authors suggesting with this statement that region of origin could be used as a screening tool to select candidates for follow-up FADS genotyping (e.g., Mexicans but not Puerto Ricans should be screened)? Without further explanation, the idea of self-identified ancestry as a translational tool for FADS function sounds far-fetched. In contrast, the genotype of rs174537 alone would be cheap and substantially more accurate.

Response: We thank the reviewer for these insightful comments. As shown in **Table 1**, the mean proportions of Amerind ancestry vary greatly by self-identified regions of origin, with Mexican, Central American and South American Hispanics showing the greatest proportions, and individuals identifying as Cuban, Dominican and Puerto Rican showing considerably lower proportions. While anecdotal evidence suggests some Hispanic Americans may have more refined knowledge of their ancestry based on family histories, we are not aware of prior research documenting this phenomenon in greater detail. Further, while we agree with the Reviewer that it would be ideal to include genotyping of rs174537 in routine practice, current health care practice does not provide adequate resources to genotype all individuals. Thus, a priori information that individuals may have on their ancestry may serve as a preliminary tool to prioritize those who would benefit from additional screening either through genotyping or screening for n-3 LC-PUFA deficiency. We have now updated our Discussion (p. 18, lines 430-440) to clarify these points:

“Prior research demonstrates that mean proportions of Amerind ancestry vary greatly by self-identified regions of origin among Hispanic Americans, with Mexican, Central American and South American Hispanics showing the greatest proportions, and individuals identifying as Cuban, Dominican and Puerto Rican showing considerably lower proportions.^{61,96} While a long term goal of applying precision nutrition may include genotyping of rs174537 (or related FADS region variants)_in routine health care screening, current health care practice does not provide adequate resources to genotype most individuals. Thus, a priori information predictive of ancestry such as country or origin or otherwise, may serve as a preliminary tool to prioritize those who are most likely to have low circulating and tissue levels of n-3 LC-PUFA and would benefit from additional screening either through genotyping or screening for n-3 LC-PUFA deficiency.”

Minor comments:

1. I would suggest that the authors add body mass index. This is a more interpretable cardiometabolic factor compared to individual height and weight measurements.

Responses: We agree and have added results for BMI to the paper in the same sections where other cardiometabolic factors are presented (see **Table 2**).

2. Figure 3 and S2 are hard to interpret: please provide the sample size across genotype, provide error bars for the upper panels that show mean levels by genotype (or explain why error bars are not needed), and describe what the error bars indicate for the lower panels that show the effect size (standard deviation?). For the effect sizes, it looks like the effects of TT are in reference to GG (as opposed to GT), but this needs to be explicitly stated.

Response: We added the sample size across genotype and the error bars. We have clarified what genotype was used as the reference in the legends for **Figure 3** and **Figure S4** (previously Figure S2).

Updated Figure 3: Genotypic effects of rs174537 on Triglycerides and E-selectin.

Figure 3 shows the effect of rs174537 on Triglycerides and E-selectin. The sample size across genotype is 293 for GG, 484 for GT, 325 for TT. Upper figure shows the mean and standard deviation of Triglycerides and E-selectin stratified by rs174537 genotypes and lower figure shows the estimated effect and standard error of carrying one or two copies of the ancestral allele (compared to the reference of zero), after covariate-adjustment for age and sex.

Updated Figure S4: Genotypic effects of rs174537 on triglycerides and E-selectin.

Figure S4 shows the effect of rs174537 on triglycerides and E-selectin. The sample size across genotype is 293 for GG, 484 for GT, 325 for TT. Upper figure shows the mean and standard deviation of Triglycerides and E-selectin stratified by the genotypes of rs174537. Center figure shows the estimated effect and standard error of carrying one or two copies of the ancestral allele (compared to the reference of zero). The effect-size estimates are adjusted for age and sex. Bottom figure shows the estimated effect and standard error of carrying one or two copies of the ancestral allele (compared to a reference of zero), adjusted for age, sex and principal components of ancestry.

3. The reference to “(Figure 2 -left)” at lines 279-280 is in the wrong place, I believe. It should be earlier when the unadjusted regression is described.

Response: We thank the reviewer for identifying this error and we have made the correction.

4. Table 2: the reference group for the genotypic effects (FADS GG genotype?) should be explicitly clear

Response: We apologize for the oversight. We have clarified the reference group as *FADS* rs174537 GG genotype in the current revision.

5. Self-reported fish intake was measured in MESA and appropriately adjusted for by the investigators. Is the use of fish oil or other supplements that impact PUFA levels and associated metabolites available in MESA. If not, this should also be listed as a limitation. This is partially covered under the “diet and lifestyle” limitations that the authors state, but should be more specifically mentioned. Based on the histograms in the supplement that show a small subset of participants with very high PUFA levels, it seems likely that supplements may be playing a role here.

Response: As we now note in the Supplementary Methods (p. 2), use of nutritional supplements including “Cod liver oil, other fish oils or omega-3 fatty acids” was reported through the MESA Food Frequency Questionnaires (FFQ). We used these responses to verify that none of the participants included in our analyses reported use of these supplements. We have also added to the Discussion (p. 17, lines 405-409) to highlight this point and to note possible limitations of the self-reported supplement use:

“Additionally, we used food frequency questionnaire data to confirm participants included in our analyses did not have self-reported use of fish oil supplements, and we performed analyses adjusted for self-reported fish intake in MESA. Still, we recognize there are inherent limitations with the quality of self-report-based measures of diet and supplement use.”

REVIEWERS' COMMENTS:

Reviewer #1 (Remarks to the Author):

The authors have carefully addressed all my comments. I have no more comments.

Reviewer #2 (Remarks to the Author):

In their response to the reviewers comment and the revised version, the authors have adressed the major problems of the original contribution. While there are remaining issue to be resolved (maybe in future papers), I find that the results of the additional analyses that they performed and their revision of the text have significantly improved the paper, to the point where I have no major objections to its publication

Reviewer #3 (Remarks to the Author):

The authors have adequately addressed previous comments.